# Spatial parameters associated with the risk of banana bunchy top disease in smallholder systems

Kéladomé Maturin Géoffroy Dato[1]*, Mahougnon Robinson Dégbègni[1], Mintodê Nicodème Atchadé[1,2], Martine Zandjanakou Tachin[3], Mahouton Norbert Hounkonnou[1], Bonaventure Aman Omondi[4]

1 International Chair in Mathematical Physics and Applications (ICMPA-UNESCO CHAIR), University of Abomey-Calavi, Cotonou, Benin Republic, 2 National Higher School of Mathematics Genius and Modelization, National University of Sciences, Technologies, Engineering and Mathematics, Abomey, Republic of Benin, 3 Doctorate School of Agriculture and Water Sciences, National University of Agriculture, Kétou, Benin Republic, 4 Bioversity International (Alliance of Bioversity International and CIAT), Abomey Calavi, Cotonou, Benin

* dageomark2@yahoo.fr

**Data Availability Statement:** All data used in this article is available on this URLs: https://figshare.com/articles/dataset/BBTD_data_Benin_used_for_

## Abstract

The Banana Bunchy Top Disease (BBTD), caused by the Banana Bunchy Top Virus (BBTV) is the most important and devastating in many tropical countries. BBTD epidemiology has been little studied, mixed landscape smallholder systems. The relative risks associated with this disease vary between geographical areas and landscapes. This work analyzed the management and vegetation conditions in smallholder gardens to assess the factors linked to landscape-level BBTV transmission and management. Mapping was done in this study area which is in a BBTD-endemic region, involving farmers actively managing the disease, but with household-level decision making. A spatial scanning statistic was used to detect and identify spatial groups at the 5% significance threshold, and a Poisson regression model was used to explore propagation vectors and the effect of surrounding vegetation and crop diversity. Spatial groups with high relative risk were identified in three communities, Dangbo, Houéyogbé, and Adjarra. Significant associations emerged between the BBTD prevalence and some crop diversity, seed systems, and BBTD management linked factors. The identified factors form important candidate management options for the detailed assessment of landscape-scale BBTD management in smallholder communities.

## Introduction

Cultivated in more than 150 countries, *Musa* spp (banana and plantain—hereafter referred to as "banana") is now the fourth-largest food crop in the world, after rice, wheat, and maize [1]. It is a key staple food for millions of people and plays an important role in the social and economic security of many rural communities. In many parts of Africa, the banana is the ultimate food crop. Banana is consumed ripe (dessert), cooked, or processed into snacks, soft drinks and fermented beverages. Plantain is mostly starchy and cooked before consumption, but

Poisson_model_and_Spatial_scan_statistics_
analysis/16743991.

**Funding:** This work was supported, in whole or in part, by the Bill & Melinda Gates Foundation [BBTV mitigation: Community management in Nigeria, and screening wild banana progenitors for resistance. Project: INV-010652 (formerly OPP1130226)]. Under the grant conditions of the Foundation, a Creative Commons Attribution 4.0 Generic License has already been assigned to the Author Accepted Manuscript version that might arise from this submission. We also have this work was funded by the University of Queensland Grant to Bioversity International under the project: BBTV mitigation: Community management in Nigeria and screening wild banana progenitors for resistance – Remote Mapping and Detection of Banana Canopy in Mixed Landscapes (Number 1229). Community Management of BBTD was funded by the Consortium Research Programme on Root Tubers and Bananas, BA 3.4, Banana Viral Diseases. We are grateful to RTB and her donor consortium. The funders had no role in study design, data collection and analysis, decision to publish, or preparation of the manuscript.

**Competing interests:** The authors have declared that no competing interests exist.

cooking banana varieties exist also. Banana plant parts and gardens also have other domestic, and cultural uses [2]. Banana/plantains are important as a cash crop, sometimes the sole source of income for rural people [3]. In central Africa, banana value chains employ more than 50% of the rural working population and contribute significantly to the GDP [4]. In Benin, bananas and plantains are among the most produced, consumed, and traded commodities [5]. Bananas and plantains are mainly grown in the southern part of the country. Due to inadequate water availability in Benin, banana is typically grown in low-lying areas, prone to flooding but providing longer water availability, unlike the highland regions elsewhere [5].

The yield gaps in banana production, particularly in smallholder systems in Africa, are increasingly being attributed to the attacks of pests and diseases, which reduce growth, the number of productive plants, or the quality of the fruit [6]. Although hitherto considered largely pest, free, bananas have recently come under increasing biotic pressure [7]. These have been attributed to more intense and widespread cultivation making croplands more connected, dominated by monocultures, and to landscapes dominated by fewer varieties. The key biotic challenges include fungal diseases, bacterial diseases, nematodes, insect pests, and plant viruses [8–10]. The banana bunchy top disease (BBTD), caused by the banana bunchy top virus (BBTV), is devastating in many tropical production systems. It is transmitted by the planting of infected planting materials, and the banana aphid (*Pentalonia nigronervosa*). BBTD causes a progressive loss of production and a simultaneous decrease in the availability of clean planting materials for the rehabilitation of infested gardens. Where farmer seed exchange dominates, it has been associated with diversity loss of farmer landraces [11]. In Benin, banana production had decreased from 18,963 tons in 2013 to 16,504 tons in 2016 since BBTD was detected in 2012; representing a 13% drop [3].

Invasive species are a particular threat to agricultural production, native biodiversity, and the stability of ecological functioning [12]. New pest introductions are increasingly common due to human and environmental factors [13]. Other factors such as the presence of susceptible hosts, connectivity between potential hosts in the landscape, and the effectiveness of dispersal mechanisms, determine the establishment and spread of these pests and pathogens. Invasive pests pose a substantial threat to sustainable banana production in Africa, with a significant risk to destabilize food security and household income in this region [14]. Banana bunchy top disease is one of the most important diseases of bananas, causing severe crop losses in many banana-growing regions [15–17]. It is rapidly spreading in the banana production belt of Benin, since being reported in 2012, in Ouémé, near the Benin/Nigeria Border [18] and later on the Nigerian side of the border [19]. The source of these outbreaks is not certain, as the closest outbreak was in southern Cameroon, Gabon, and the Democratic Republic of Congo (DRC) in central Africa. Banana bunchy top disease spreads although two ways: primary infestation through incoming infected planting material and secondary local transmission between farms by vector aphids and local seed sharing. Thus, within an endemic region, BBTD infection risk at the plot level depends on local factors influencing vector movement and seed movement. Banana production would still be possible within the BBTD affected areas if strict protocols were followed [20]. However, reducing the risk at the landscape level is desirable to make reinfection less likely. An understanding of the landscape scale BBTD risk is important in coordinating community recovery efforts.

Remote sensing is increasingly becoming a practical tool in crop management. It involves obtaining information through the analysis of images acquired using a device that does not make physical contact with the object studied [21]. Conventionally, it has been associated with satellites and aircrafts equipped with different sensors [22], and more recently unmanned aerial systems (UAS) technology [23,24]. The use of UAS to monitor crops offers great possibilities to acquire field data in an easy, fast, and cost-effective way [25]. Indeed, UAS is now

one of the most powerful tools in precision agriculture [26]. This approach involves data collection, field variability mapping through the use and development of algorithms, and decision-making to inform management practice [23–27].

Smallholder farmers produce bananas in variable landscapes and crop associations. Banana is cultivated in monoculture plantations in mixtures of varieties often of different ages. Banana association with annual or perennial crop and tree species in cleared primary and secondary forests, and in backyard gardens close to residential areas is common. Thus, the banana canopy is presented to remote scanning tools in variable backgrounds in which tools for mapping species and diseases are still poorly developed. Recently, machine learning protocols involving a first step of identifying bananas accurately in a complex matrix of crops, before narrowing down to disease symptoms have been attempted [23]. This protocol is, however, still limited to the detection of advanced disease symptoms. Ground observations and the assessment of production history are therefore important in determining the underlying disease risk, support recommendations for further manual monitoring, and supporting community management and collaboration for local production recovery.

Studies on BBTD epidemiology are mostly based on subtropical large-scale monocultures [28,29]. Some of these studies have yielded a useful framework for BBTD management in smallholder systems in Africa [20]. [30] designed a model for understanding BBTD prevalence using hypothetical model farms, based on distance from the point of infection. Recently, [28] have modelled the spread of BBTD in sections of a plantation. These studies reveal important aspects of BBTD spread, but do not consider the landscape diversity in smallholder production systems. This study, therefore, investigates the risk factors associated with BBTD prevalence in mixed canopy smallholder production systems. We used UAS, manual ground-truthing, and farmer surveys to take spatial information, such as banana mats location and vegetation diversity, for BBTD risk determination. A ground-truthing assessment was done in parallel to assess BBTD prevalence around individual gardens.

The following sections are arranged as follows: in "Materials" and "Methods", the materials and the methods are briefly presented. "Results" describes the outputs of the analysis. Finally, in "Discussion" we discuss the work, and future studies generated from this study.

## Materials and methods

### Study area

This study was carried out in six municipalities in southern Benin: Akpro-Missérété, Adjarra, Dangbo, Sakété, Athiémé, and Houéyogbé, representing the main banana-growing regions, where BBTD has been reported (Fig 1). These fall in the departments of Ouémé, Plateau, Mono, respectively. Banana and plantain in Benin are mainly grown in the humid south, a region characterized by warm temperatures (27–32˚C) and bimodal rainfall with peaks in April/ May and September and totalling between 1000 and 1300 mm p.a. This study was carried out from July to August 2019, representing the period of high rainfall during the main rainy season. This represents a period after three months of vigorous growth of banana plants during the rainy season.

Authorisation for UAV flight within the territory of the Republic of Benin operation was granted following an application to the Ministry of Agriculture and military council of the presidency (letter attached). Local administration was further informed of project activities, as part of collaborator work with a local university. Farmers who were part of the data collection gave verbal consent for flights and survey (see survey document for wording), with the first demonstration of drone operation being done in a common demonstration garden before the survey was started.

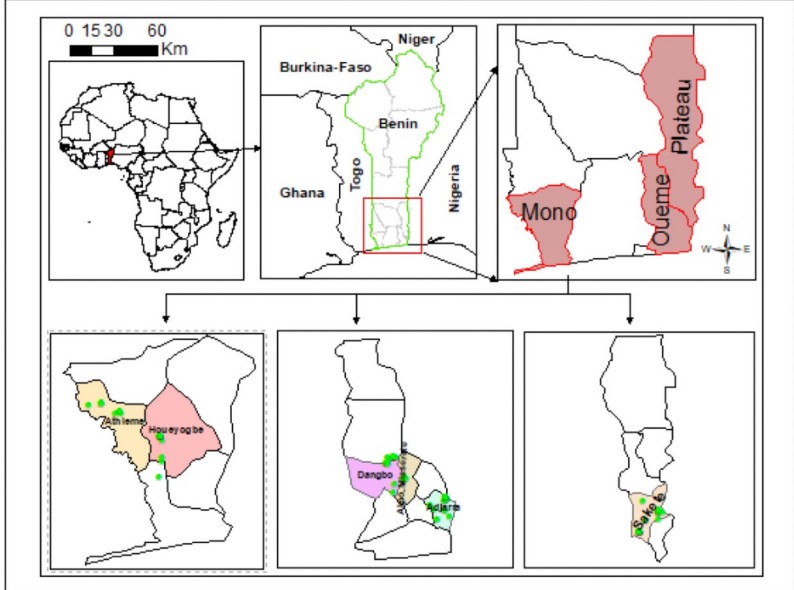

**Fig 1. Location of study sites showing banana gardens.** (green dots) Banana fields surveyed.

Seventy-one banana gardens in six municipalities were surveyed. A banana garden was considered as a continuous unit with discernible limits, under a single owner, or planted in the same season. These farmers are participants in the BBTD recovery project, hence assumed capable of recognizing most symptoms of BBTD. Their gardens were also assumed to closely represent the banana production systems in this region. Within each municipality, all farmers available for a ground-truthing interview also had their farms surveyed using the UAS. Thus, we created a dataset of crop scouting and survey with the management history for each plot. A banana plot was the unit of the survey, thus where a farmer had more than one plot, each was treated as a separate entity.

## Image capture and analysis

The protocol for data collection and analysis is summarized in Fig 2. Images were taken between 10 am and 5 pm, in a clear, calm (cloud cover < 20% and wind speeds between 7 and 34 km/h) using a Phantom 4 Pro and Phantom 4 Pro+ (DJI, China) UAV (Unmanned Aerial Vehicle), equipped with a 1-inch, 20-megapixel sensor multimedia camera. The Phantom 4 Pro+ is capable of recording in Ultra High Definition (UHD) 4K (4096x2160) at 60 fps (feet per second), at a maximum bit rate of 100 Mbps. For data collection, mission planning was done using PIX4D CAPTURE Software (4.5 version). At each garden, the UAV was flown at an altitude of 40 meters, allowing a footprint of approximately 5472 × 3648 pixels with a spatial resolution of 7 cm. The flight mission was designed to leave target plots at the centre of the area surveyed with a buffer of up to 100m. The minimum length of flight area flow is 75m ×75 m, about 5.6 ha, and the minimum number of photos obtained is 230 (Fig 3). The camera was programmed to take a picture every two seconds, ensuring 70–80% overlap between each pair of adjacent images. Thus, every single point was covered at least four times during the mission. All flights maintained the line of sight of the operator to the UAV. Airspace regulations as defined in Benin by the Civil Aviation Authority were followed, and no restricted areas were overflown. The flight time at each site was about 20 minutes and the pattern used is the grid

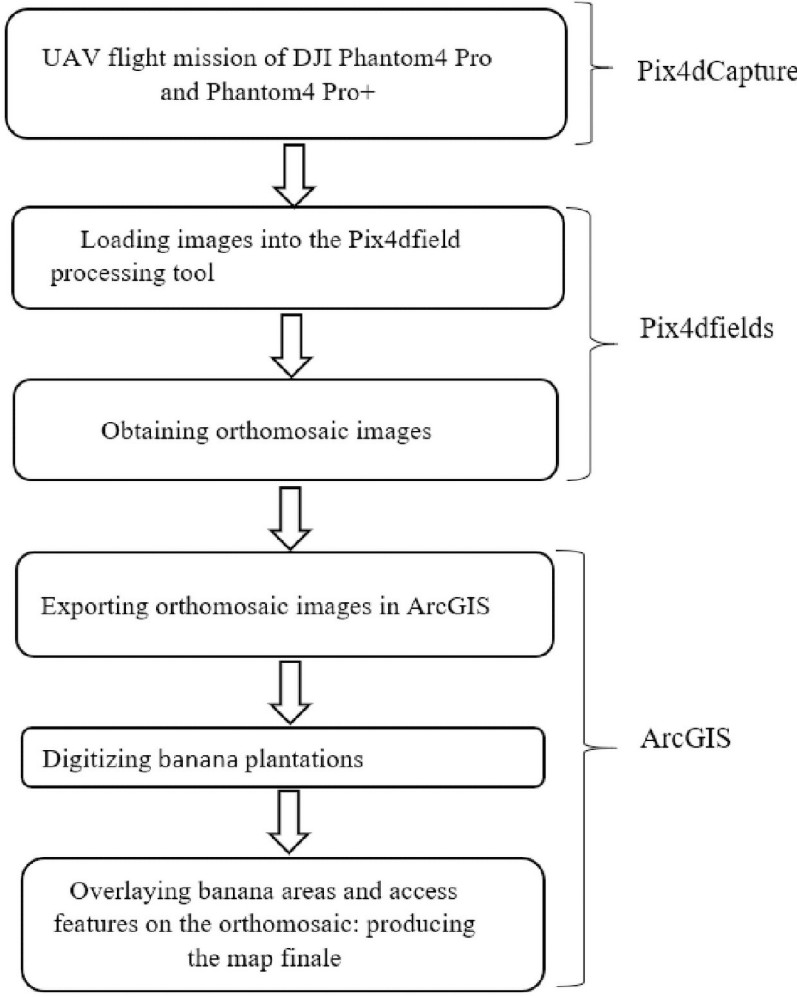

**Fig 2. Summary of the study analysis workflow for image processing in Pix4Dfields and ArcGIS.**

mission. A safety margin of 10 minutes of flight representing 30% of battery power was left for safe landing and take-off manoeuvrers.

The images collected were automatically geotagged by the onboard GPS, so the orthomosaic could be georeferenced without the need for image, map, or ground control point (GCP) references. The resulting image from the flights was imported into PIX4DFIELDS (1.5 version, PIX4D) and processed to extract an orthomosaic including a correction for topographical distortions, for each plot following a standard procedure: (i) aligning the photographs (accuracy: high; alignment: reference); (ii) constructing a dense point cloud (quality: high; depth filtering: moderate); (iii) constructing a digital elevation model (DEM) (pixel size 7 cm; interpolation: extrapolation; all classes of points to be generated digital surface model); (iv) constructing an orthomosaic (input area: DEM; mixing mode: mosaic) (See S1 Fig). Once processed, the resulting orthomosaic was imported into ARCGIS Desktop (10.7.1 version) to digitize banana mats (S2 Fig).

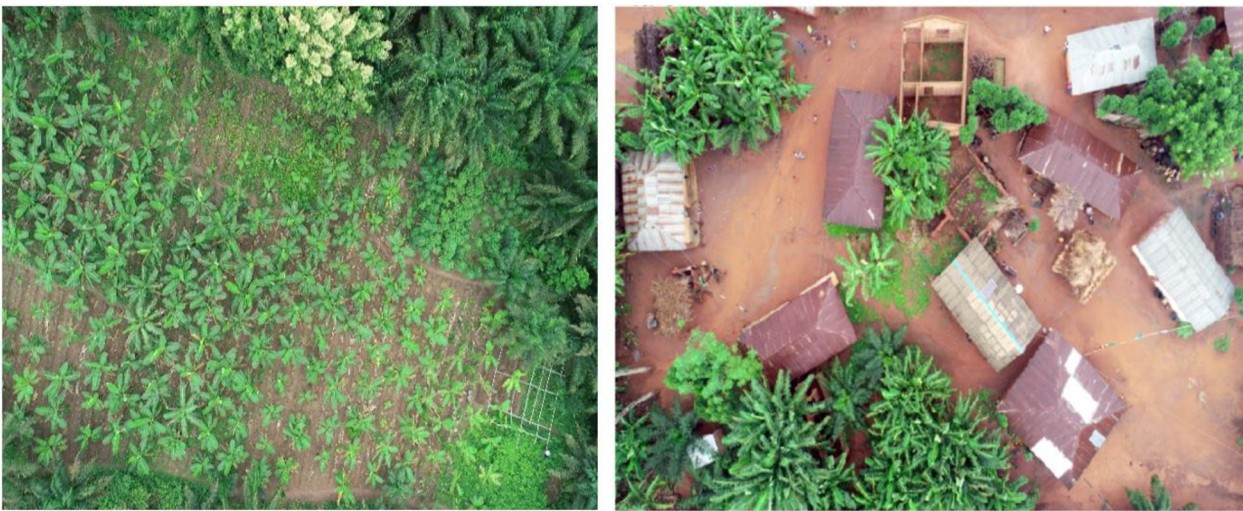

**Fig 3. Photo took by the UAV during flight missions.** (left) Open gardens. (Right) backyard gardens.

## Field ground-truthing survey

Parallel to the UAV study, a ground-truthing survey was conducted in each field to observe selected environmental variables. We recorded the presence and type of vegetation around the target field, wind direction and speed, banana tree density, and types of banana mats used. A survey ($n$ = 71) questionnaire was used to assess the crop and BBTD management history of each plot. These observations were made within and around it at 30m, 60m, and 90m radius from the target plot. Brief historical information of the garden (age of the plot and cropping management) was sought from the proprietors. For all analyses, the response variable was the BBTD symptomatic plants observed, assuming that the number of plants showing BBTD closely estimated the actual number of plants infected with BBTV [20].

## Data processing and analysis

The survey forms were encoded in the CSPRO software (7.2 version on www.census.gov), designed beforehand, and summarized for analysis. The database was imported into SATSCAN software (9.6 version, https://www.satscan.org/) for further analysis. SATSCAN was used to detect spatial groups and test if they were statistically significant through purely spatial approaches using discrete scan statistics of Poisson and with high- or low-rate scans for unique risk-associated attributes of geographical areas around the sampling points. A generalized linear model was done in R (3.6 version) following the procedure of Poisson regression [31].

Spatial modelling was done as follows.

The notations adopted for the presentation of this method were:

$G$: the study area (all the six municipalities)

$N$: the number of Spatial Units (SU) within the study area (all the 71 fields)

$P$: the total number of individuals (population at risk: total number of banana mats in the study area)

$P_i$: the number of individuals in the SU $i$ for $\forall i$, $i \in 1, \cdots, N$ (total number of banana mats in each field)

$Z$: the disc of current interest and a collection $\Omega$ of $m$ possible discs $Z$ such as $\Omega = \{Z_i \subset G \mid i = 1, \cdots, m\}$

$P_{z_i}$: the number of individuals in the area $Z_i$

$C$: the random variable representing the total number of cases in the study area

$C_i$: the random variable representing the number of cases in the SU $i$ for $\forall i, i \in 1, \cdots, N$

$c$: the total number of cases observed in the study area

$c_i$: the number of cases observed in the SU $i$ for $\forall i, i \in 1, \cdots, N$

$c_{z_i}$: the number of cases among $P_{z_i}$

## Purely spatial scanning statistics

The spatial scan statistic [32] was used to identify areas with abnormal BBTD incidences that were least consistent with the null hypothesis of constant risk within the study area. This method is based on a likelihood ratio test and tests the hypothesis that disease risk remains constant within the study area. It is very powerful and can be applied to both grouped and single data points [32]. A window of predefined shape (disk or ellipses), of variable size, scans the study area. For each window, a statistic, based on the likelihood ratio and the number of observed and expected cases, was calculated. The likelihood functions follow the theoretical distribution associated with the number of cases. Two distribution models were used: the Poisson's law (aggregate data or when the number of cases is negligible compared to the size of the population) and the binomial law (individual case and control data). The window that corresponded to the maximum likelihood was the most likely group, the one least likely to occur by chance. To test the significance of the groups detected, Monte Carlo simulations are performed (with up to 999 replications).

## Detection and identification of BBTD groups

Spatial scan statistics, implemented in the SATSCAN software, were used to test for the presence of BBTD spatial groups and to identify their approximate locations. Group evaluation was performed by comparing the number of cases within the window with the expected number if the cases were randomly distributed spatially. The test of significance of the identified groups was based on a likelihood test whose $p$-value was obtained by the Monte Carlo test [33]. The identification of spatial groups with high or low rates was carried out under the assumption of the Poisson probability model, using a maximum spatial group size of 5% of the total population. For statistical inference, 999 Monte Carlo replications were performed. The null hypothesis of no group was rejected when the simulated $p$-value is less than or equal to 0.05 for the primary group and 0.10 for the secondary group.

[34] defined a spatial scanning statistic that imposes a circular window and allows the centre of the disk to move across the map. For any given position of the centroid, the radius of the disk varies continuously from zero to a predefined upper limit ($100m$). Theoretically, the method uses an infinite number of separate disks, each with a different location and size. For the analysis, we define the disc as comprising no more than 5% of the total population. In the alternative hypothesis, there is at least one disk with a higher relative risk on the inside compared to the outside. For each disk, it is possible to calculate a likelihood that takes into account the number of cases observed inside and outside the disk. The disk that maximizes the likelihood function is called the most likely group or the candidate group. The most common

disease mapping applications take into account the Poisson distribution or the binomial distribution of the number of disease cases. The Poisson model was defined as follows:

Considering the map $G$, as our study area, and $Z$, the disc of current interest and a collection $\Omega$ of $m$ possible discs $Z$ such as $\Omega = \{Z_i \subset G \mid i = 1, \cdots, m\}$ [34,35].

$P_{z_i}$ is the number of individuals in a zone $Z_i$ and $C_{z_i}$ is the number of cases among $P_{z_i}$. Let $A$ be a zone included in $G$, $P_A$ the number of individuals in $A$, and $C_A$ the random variable of the number of cases in $A$. For $\forall A \subset G$, $C_A$ is generated by an Inhomogeneous Poisson Process (IPP) such as:

$$C_A \sim Poisson(\pi P_{A \cap Z_i} + \delta P_{A \cap \bar{Z}_i})$$

where $\pi$ is the probability of being a case in $Z_i$, $P_{A \cap Z_i}$ is the number of individuals at the intersection of zones $A$ and $Z_i$, $\delta$ is the probability of being a case outside $Z_i$, $P_{A \cap \bar{Z}_i}$ the number of individuals in zone $A$ and outside zone $Z_i$. Under the null hypothesis ($H_0$) of risk homogeneity: $\pi = \delta$ et $C_A \sim Poisson(\pi P_A) \forall A$. Under the alternative hypothesis ($H_1$): $\pi > \delta$.

a: <u>Calculation of the likelihood function for all zones $Z_i$.</u>

In the region $G$, knowing the $Z_i$ window, the probability of observing $c$ case is such that:

$$Prob(c) = \frac{e^{-(\pi P_{z_i} + \delta(P - P_{z_i}))} \times (\pi P_{z_i} + \delta(P - P_{z_i}))^c}{c!} \tag{2.1}$$

Each case $x$ of the $c$ cases has a probability of being in a given area of $G$ which depends on the population at risk in that area.

The function $f(x)$ of the probability density of the cases in $G$ known $Z_i$ was defined such that:

$$f(x) = \frac{\pi P_x}{\pi P_{z_i} + \delta(P - P_{z_i})} \times I(x \in Z_i) + \frac{\delta P_x}{\pi P_{z_i} + \delta(P - P_{z_i})} \times I(x \in \bar{Z}_i) \tag{2.2}$$

$I(x \in Z_i)$ and $I(x \in \bar{Z}_i)$ are being indicators whose value is equal to 1 if $x \in Z_i$ (respectively $x \in \bar{Z}_i$) at 0 if not.

The likelihood function for the $Z_i$ window is equal to:

$$L(Z_i, \pi, \delta) = Prob(c) \times \prod_{j=1}^{c} f(x_j) \tag{2.3}$$

By developing this equation, we obtain:

$$L(Z_i, \pi, \delta) = \frac{e^{-\lambda}}{c!} \times \pi^{c_{z_i}} \delta^{(c - c_{z_i})} \times \prod_{j=1}^{c} P_{x_j} \tag{2.4}$$

with $\lambda = \pi P_{z_i} + \delta(P - P_{z_i})$

b: <u>find the $Z_i$ zone that maximizes the likelihood function.</u>

To do this, we ask:

When $\frac{c_{z_i}}{P_{z_i}} > \frac{c - c_{z_i}}{P - P_{z_i}}$,

$$L(Z) \overset{def}{=} \sup L(Z_i, \pi, \delta) = \frac{e^{-c}}{c!} \times \left(\frac{c_{z_i}}{P_{z_i}}\right)^{c_{z_i}} \left(\frac{c - c_{z_i}}{P - P_{z_i}}\right)^{c - c_{z_i}} \times \prod_{j=1}^{c} P_{x_j} \tag{2.5}$$

Otherwise

$$L(Z) = \frac{e^{-c}}{c!} \times \left(\frac{c}{P}\right)^c \times \prod_{j=1}^{c} P_{x_j} \tag{2.6}$$

The maximum likelihood estimator $\hat{Z}$ is such that:

$$\hat{Z} = \{Z_s \in \Omega : L(Z_s) \geq L(Z_i) \,\forall\, Z_i \in \Omega\} \tag{2.7}$$

The likelihood ratio is:

$$LR = \frac{L(\hat{Z})}{L_0}$$

With

$$L_0 \overset{def}{=} \underset{\text{p=q}}{sup}\ L(Z_i, \pi, \delta) = \frac{e^{-c}}{c!} \times \left(\frac{c_{z_i}}{P_{z_i}}\right)^{c_{z_i}} \left(\frac{c - c_{z_i}}{P - P_{z_i}}\right)^{c - c_{z_i}} \times \prod_{j=1}^{c} P_{x_j}$$

#### c: Inference

The significance of the model was tested by Monte Carlo inference, as the distribution of the statistic is not known. The most likely groups were obtained at the 5% significance level, while the secondary groups are determined at the 10% threshold. This zone is the $Z_s$ window (defined in (2.7)) which maximizes the likelihood function and is, therefore, the most likely group.

## Relative risk

The real relative risk ($RR$) is defined as the risk $\lambda_z$ in the zone $Z$ compared to the risk $\lambda_{(G\setminus Z)}$ in all other regions $G \mid Z$:

$$\lambda_Z = \frac{E(C_Z)}{E_Z}, \tag{2.8}$$

$$E_Z = c \frac{P_z}{P} \tag{2.9}$$

where $C_Z$ is the Poisson random variable representing the number of cases in the region $Z$, with the expected value given by $P_Z$ the population of the region $Z$, $P$ is the total population of the map $G$, $E_Z$ is the expected number of cases in region $Z$ under the null hypothesis, and $c$ is the total number of observed cases. Similarly, we define $\lambda_{(G|Z)}$. In this way, the true relative risk is defined as follows:

$$RR = \frac{\lambda_Z}{\lambda_{(G|Z)}}$$

If $Z$ and $G \mid Z$ both have the same $\lambda_Z = \lambda_{(G|Z)} = \lambda$, the real relative risk is 1. Suppose that $Z$ is selected independently of the observed data. The natural estimator of $RR$ is given by:

$$\widehat{RR} = \frac{c_Z/E_Z}{(c - c_Z)/(E_G - E_Z)} \tag{2.10}$$

where $c$ is the total number of cases, $c_Z$ is the number of cases in the group $Z$, $E_G$ is the expected number of cases in the region under the null hypothesis, and $E_Z$ is the expected number of

cases in the region $Z$ under the null hypothesis. This is an unbiased estimator when the region is chosen independently of the observed data. Thus, if the estimated risk in the $Z$ region, $c_Z/E_Z$, is close to the estimated risk outside the group $(c - c_Z)/(E_G - E_Z)$, we have $\widehat{RR}$ which is close to 1 and there is strong evidence that there is no group on the map $G$. If $c_Z/E_Z$ increases relative to $(c - c_Z)/(E_G - E_Z)$, the estimated relative risk increases by indicating the existence of a group in $Z$ region. However, we are only interested in estimating the $RR$ for the candidate group. Because $\hat{Z}$ is chosen among all possible $Z$ to maximize (2.5), it is highly dependent on the random variables realized.

The GLM model was run to determine the variables most significantly associated with the response variable (BBTD prevalence) in the examined gardens and their buffer areas. GLM enabled the use of both continuous and categorical variables (Table 1). GLM was run in R software (3.6 version) assuming a Poisson distribution, with a significance threshold of 5%. However, the variables showing a 10% limit were noted. The model was validated using the Hosmer-Lemeshow test and Cameron and Trivedi test. The Hosmer-Lemeshow test was used to assess the relevance of the Poisson regression model. The associated $p$-value uses the Chi-square distribution, and so the model was assumed to be adapted to the data if the $p$-value obtained was $1 > 0.05$. The Cameron and Trivedi test were used to assess the abnormality of the dispersion of the response variable. The associated $p$-value obtained is equal to $0.9987 > 0.05$. Thus, we conclude that there is a normal dispersion of $Y$ values represented by the number of cases of BBTD.

## Results

Spatial clustering analysis of all 71 fields identified three statistically significant high-risk areas, covering a total of 21 fields, distributed in all six communes (Table 2). The rest of the 50 fields assessed did not have a BBTV infection risk significantly different from the general background. The total number of fields with reported cases is 64 and the total number of reported cases (mats showing BBTD) is 168. A total of 30 groups were detected, 7 of which were significant at the 5% threshold and 11 significant at the 1% threshold. The shapefile of the groups generated was exported and a Keyhole Markup Language (KML) file in GOOGLE EARTH, and projected onto a map to identify the locations of the clusters on the ground. Most of the clusters were detected in Dangbo (13 clusters, 15 fields, 20.4 mean risk factor) and the lowest was detected in Athiémé and Sakété. Results suggest the existence of BBTD "high-risk areas" in certain regions, namely, Dangbo, Houéyogbé, and Adjarra showed relative higher risk and representation of fields with high relative risk compared to Akpro-Missérété, Athiémé, and Sakété (S3 Fig).

### The output of Poisson regression

The results of the Poisson regression show 11 significant explanatory variables at the 5% threshold. Crop association with banana was associated with lower BBTD occurrence. In General, 8 crops were found in association with banana/plantains spanning different growth habits annuals, perennials, biennials, and use classes—grains, legumes, and horticultural crops. The more crops are associated, the lower the number of cases of BBTD were observed. Banana monocrops thus had the greatest risk of BBTD in all communes (S4 Fig). Among the individual crop types, fruits crops, and other crops (sugar cane, legumes, taros, and palm oil) had the greatest association with BBTD, while maize and cassava had the lowest. Banana crop preceding the current stand (continuous/perennial banana cropping) gave the greatest risk of BBTD, similar to tomato crops ($n = 3$, $p$-value = 0.0069); while maize and cassava preceding the banana crops had the lowest association with BBTD risk ($n = 7$, $p$-value = 0.1087). The plots

**Table 1. Presentation of Poisson regression variables.**

| Variable | Description | Modalities | N | Type |
|---|---|---|---|---|
| Q13 | Number of BBTD Cases | - | 39 | Quantitative |
| QA | Locations | Akpro-Missérété | 4 | Qualitative |
| | | Dangbo | 14 | |
| | | Sakété | 16 | |
| | | Houéyogbé | 11 | |
| | | Adjarra | 19 | |
| | | Athiémé | 7 | |
| Q1 | Cultivation system | Backyard gardens | 17 | Qualitative |
| | | Open gardens | 54 | |
| Q2 | Crop on plot 1 year before current banana crop | None (Fallow) | 4 | Qualitative |
| | | Maize | 6 | |
| | | Cassava | 1 | |
| | | Banana | 52 | |
| | | Tomatoes | 3 | |
| | | Others (vegetable) | 2 | |
| | | Others (taro) | 1 | |
| | | Others (palm oil) | 1 | |
| | | Others (peanut) | 1 | |
| Q5 | Type of *Musa* | Plantain | 17 | Qualitative |
| | | Sweet banana alone | 1 | |
| | | Plantain/sweet banana mixture | 53 | |
| Q7 | Type of associated crop | Maize (cereals,annual) | 7 | Qualitative |
| | | Beans (legume) | 1 | |
| | | Cassava | 3 | |
| | | Yam | 1 | |
| | | Fruit crops | 2 | |
| | | Tomatoes (vegetable) | 3 | |
| | | Others (vegetable) | 6 | |
| | | Others (taro) | 7 | |
| | | Others (palm oil) | 7 | |
| | | Others (cacao) | 1 | |
| | | Others (sugar cane) | 5 | |
| | | Others (laurel) | 2 | |
| | | Pure banana (monocrop) | 26 | |
| Q8 | Type of planting material | Suckers | 56 | Qualitative |
| | | Macropropagation | 10 | |
| | | Suckers and Macropropagation | 4 | |
| | | Suckers, Macropropagation and Vitroplants | 1 | |
| Q9 | Source of planting material | NGOs | 2 | Qualitative |
| | | Friends | 43 | |
| | | Institutes | 6 | |
| | | Own field | 20 | |
| Q14 | Presence of aphids on the mother plant | No | 1 | Qualitative |
| | | Yes | 70 | |
| Q17 | Presence of aphids on the suckers | No | 8 | Qualitative |
| | | Yes | 63 | |

(*Continued*)

**Table 1.** (Continued)

| Variable | Description | Modalities | N | Type |
|---|---|---|---|---|
| Q23 | During the last planting season, did you receive planting material from neighbours? | No | 49 | Qualitative |
| | | Yes | 22 | |
| Q25 | Do you practice fallowing before a new banana plantation? | No | 67 | Qualitative |
| | | Yes | 4 | |
| Q30 | Did you destroy any existing banana mats on this plot? | No | 64 | Qualitative |
| | | Yes | 7 | |
| Q35 | Do you cut or uproot BBTD-infected banana mats? | No | 21 | Qualitative |
| | | Yes | 50 | |
| Q43.East | Type of vegetation in the immediate area | Seasonal crop (these are annual) | 16 | Qualitative |
| | | Annual crops (split into cereals and legumes and vegetables) | 1 | |
| | | Perennial crops | 29 | |
| | | Fruit trees (these are perennial) | 14 | |
| | | Forestry (Non-Fruit) | 5 | |
| | | No vegetation | 6 | |
| Q43.South | Type of vegetation in the immediate area | Seasonal crop (these are annual) | 11 | Qualitative |
| | | Perennial crops | 21 | |
| | | Fruit trees (these are perennial) | 20 | |
| | | Forestry (Non-Fruit) | 10 | |
| | | No vegetation | 9 | |
| Q43.West | Type of vegetation in the immediate area | Seasonal crop (these are annual) | 18 | Qualitative |
| | | Perennial crops | 19 | |
| | | Fruit trees (these are perennial) | 14 | |
| | | Forestry (Non-Fruit) | 8 | |
| | | No vegetation | 12 | |
| Q43. North | Type of vegetation in the immediate area | Seasonal crop (these are annual) | 8 | Qualitative |
| | | Annual crops (split into cereals and legumes and vegetables) | 6 | |
| | | Perennial crops | 17 | |
| | | Fruit trees (these are perennial) | 18 | |
| | | Forestry (Non-Fruit) | 12 | |
| | | No vegetation | 10 | |

that had been planted with tomato crop one year before the current crop were 1,357 times more likely to have BBTD cases compared to those that had banana-free fallow (S5 Fig). The older the banana garden, the more the number of cases of BBTD were observed. BBTD cases increased by 1.14 times per year, keeping the other parameters constant. Older banana crops also had a higher density of stems per square meter ($r = -0.2$). The cropping density (mats/stem per square meter) was positively correlated with disease risk ($r = 0.0528$, $p$-value = 0.0367) (See S6 Fig).

Biotic environmental, and crop management parameters were also correlated with the BBTD risk observed. Vector abundance and distribution were the single most important factor associated with BBTD risk ($n = 63$, $p$-value = 0.0461). The occurrence of aphids on the main pseudostem and the suckers were both positively correlated with BBTD risk (S7 and S8 Figs). Seed sourcing was also associated with BBTD. Farmers using suckers sourced from the neighbours were 53.1 times more likely to have BBTD in their gardens than those who used their own seeds (see S9 Fig). Several standard cultural disease control options were associated with

**Table 2. Spatial clustering of BBTD in six communes, Benin, 2019.**

| Clusters | Field ID | Locations | Observed case | Expected Case | Relative risk | Report log likelihood | p-value | Radius(m) |
|---|---|---|---|---|---|---|---|---|
| 1 | 10, 8 | Dangbo | 15 | 0.55 | 29.8 | 35.757 | 0 | 0.095 |
| 2 | 6, 7 | Dangbo | 15 | 0.77 | 21.26 | 30.911 | 0 | 0 |
| 3 | 9 | Dangbo | 8 | 0.15 | 57.13 | 24.312 | 0 | 0 |
| 4 | 47, 48 | Houeyogbe | 10 | 0.66 | 16.02 | 18.091 | 0 | 0 |
| 5 | 44 | Houeyogbe | 10 | 1.1 | 9.59 | 13.399 | 0 | 0 |
| 6 | 11 | Dangbo | 5 | 0.16 | 31.86 | 12.392 | 0 | 0 |
| 7 | 64 | Dangbo | 5 | 0.22 | 23.36 | 10.898 | 0 | 0 |
| 8 | 17 | Adjarra | 5 | 0.33 | 15.56 | 8.978 | 0.0004 | 0 |
| 9 | 45 | Dangbo | 7 | 0.88 | 8.24 | 8.499 | 0.0007 | 0 |
| 10 | 49 | Athieme | 0 | 7.35 | 0 | 7.518 | 0.0024 | 0 |
| 11 | 28 | Dangbo | 3 | 0.11 | 27.71 | 7.0479 | 0.0041 | 0 |
| 12 | 14 | Dangbo | 5 | 0.59 | 8.74 | 6.351 | 0.0170 | 0 |
| 13 | 5 | Dangbo | 3 | 0.15 | 20.77 | 6.221 | 0.0210 | 0 |
| 14 | 32 | Sakete | 0 | 6.02 | 0 | 6.133 | 0.0220 | 0 |
| 15 | 1 | Dangbo | 4 | 0.43 | 9.59 | 5.422 | 0.0280 | 0 |
| 16 | 52 | Athieme | 5 | 0.73 | 6.99 | 5.379 | 0.0330 | 0 |
| 17 | 63 | Dangbo | 3 | 0.22 | 13.84 | 5.076 | 0.0450 | 0 |
| 18 | 70 | Adjarra | 3 | 0.22 | 13.84 | 5.076 | 0.0450 | 0 |
| 19 | 56 | Sakete | 2 | 0.14 | 14.49 | 3.4747 | 0.206 | 0 |
| 20 | 65 | Dangbo | 2 | 0.15 | 13.77 | 3.3794 | 0.238 | 0 |
| 21 | 37 | Sakete | 0 | 3.33 | 0 | 3.3608 | 0.271 | 0 |
| 22 | 15 | Akpro-Misserete | 0 | 2.94 | 0 | 2.9641 | 0.408 | 0 |
| 23 | 3, 42 | Dangbo | 0 | 2.57 | 0 | 2.5907 | 0.540 | 0 |
| 24 | 61 | Sakete | 0 | 2.31 | 0 | 2.3298 | 0.658 | 0 |
| 25 | 53 | Athieme | 1 | 4.58 | 0.21 | 2.0941 | 0.797 | 0 |
| 26 | 35 | Sakete | 0 | 1.69 | 0 | 1.6980 | 0.905 | 0 |
| 27 | 27 | Adjarra | 2 | 0.48 | 4.23 | 1.3493 | 0.961 | 0 |
| 28 | 62 | Sakete | 0 | 1.32 | 0 | 1.3274 | 0.969 | 0 |
| 29 | 67, 68, 66 | Adjarra | 0 | 1.18 | 0 | 1.1794 | 0.993 | 0.078 |
| 30 | 24 | Adjarra | 3 | 1.10 | 2.75 | 1.1177 | 0.998 | 0 |

reduced BBTD occurrence in the investigated fields. Farmers reporting uprooting of diseased banana mats were also 7.9 times also reported higher levels of BBTD (S10 Fig). Similarly, securing clean seed, field isolation, and intercropping were also significantly associated with reduced BBTD. The farther away the field was from an infected field, the smaller the number of cases of BBTD were reported (S11 Fig). Border vegetation of different species was also associated with reduced BBTD infested plants reported (S12 Fig).

## Discussion

The present study assessed the spatial and production factors associating with the BBTD risk in banana gardens in southern Benin. This distribution of groups in these zones would be compatible with common risk factors, namely: type of variety, associated crops, the distance between fields, presence of barriers, etc., which would have considerably reduced the number of BBTD cases observed (S13 Fig). In addition, we also identified some estimators that could be useful in deriving BBTD risk in such efforts, especially in low altitude plantain production systems. These are consistent with the risk factors previously identified by [30]. Some level of

BBTD risk was found in all regions, with some regions including Dangbo, Houéyogbé, and Adjarra, near the Benin-Nigeria border, being the riskiest, and closest to the regions of the first outbreak within the country. Banana production in Benin is localized in the southern end of the country. There is also some cultivation in the drier areas along river valleys and lowlands. The variables obtained in this study could be useful in initiating community-based approaches to BBTD control in West Africa.

The risk of BBTD infection in replanted fields is variable even within BBTD-endemic landscapes. Overall, our data show that crop succession, production, longevity, and informal seed systems to be associated with a higher risk of BBTD; while mixed cropping matrices, higher altitude, and lower surrounding BBTD pressure were linked to reduced BBTD prevalence. Within the cropping management, the diverse kind of banana production (plantain or sweet banana) differed in their BBTD prevalence and calculated risk, as did the identity of the associated crop, the distance between fields, and the presence of barriers. This study isolates some potential combinations of approaches that could help keep banana production in BBTD endemic regions [20–36]. The results of this study suggest a set of tools that could be better refined and tested alone or in combination to determine their potential in BBTD control. Analyzing BBTD reinfection risk is important in determining where to initiate the recovery of bananas in BBTV endemic regions.

Out of the 25 variables tested, eleven showed significant association with BBTD prevalence observed. They fall into three categories: environmental factors, those related to crop growth and vector movement, seed systems and crop management factors. The role of natural barriers in reducing infection levels was an interesting finding. Rows of barriers influence the movement of vectors between plots, slowing disease spread at the landscape level, similar to associated crops within the plantation. A number of these variables could be correlated or interacting with each other. For example, older farms and those with higher banana density had a higher level of BBTD. The growth habit of bananas implies that an older crop would also have more stems per mat than recently established ones in smallholder production systems, where suckers are often not thinned. Thus, the influence of crop age on BBTD prevalence may be partly due to the effect of plant density in older plots. Conversely, intercropped fields have a lower plant density compared to monocrop bananas, both resulting in lower canopy connectivity. The individual and additive effects of these measures would be interesting to test in controlled experiments, to identify priority packages for BBTD control in smallholder systems [20]. Previously, [37] had observed the role of canopy connectivity in BBTD spread at a plantation edge in Australia (subtropical systems). This study highlights the importance of inoculum source management and landscape-scale operations in the management of BBTD in smallholder systems, and the role of population and vector movement in determining BBTD reinfection at the landscape level [20–38].

The direct correlation of vector-linked variables (e.g., aphid density, banana density, and presence of barriers) with BBTD risk showed an important contribution of those factors related to vector movement in reducing BBTD risk. Although vector control is typically not done in BBTD management, except in reducing the emigration of aphids during rouging or plant disturbance practices, the role of vector movement is clear. Such practices could include intercropping with non-host species and those likely to influence vector dispersal and host findings. Vector movement might take a temporal variation dependent on the effect of weather or plant growth on the formation of winged aphids, and the risk of vector migration. Thus, such control measures may be more effective in some seasons than others, or entirely unnecessary when vector movement is minimal. The development of winged aphid forms has been associated with vector population, host quality, and environmental cues [39]. Apterous aphids form when populations rise either due to sufficient feeding opportunities or reduced direct

mortality from weather factors. This could pose a seasonal change in disease transmission risk from migrating aphids. Optimization in this way would include designing cost-effective packages suitable for specific production situations and seasons. We estimated that the presence of aphid populations could only explain 27% of the total BBTD in these communes. Therefore, other mechanisms of virus dispersion could be driving the BBTD, particularly at slow rows of barriers around the nearest fields.

In this study, GLM was used to analyze the current spatial distribution of BBTD. Spatial scan statistics do not make it possible to take into account explanatory variables in the calculation of the expected number of cases in a spatial unit. However, in epidemiology, the spatial heterogeneity of the number of cases of the disease can be attributable to many factors such as the socio-demographic characteristics of each spatial unit. These factors are taken into account through indirect standardization and by considering only qualitative variables. One possible solution lies in the use of log-linear spatial models to take into account all types of explanatory variables and, consequently, to give an estimate of the expected number of cases per spatial unit. Testing the limits of this approach in epidemiological modelling is beyond the scope of this work, but would be interesting especially as applied to disease risk mapping or estimation of production suitability in agricultural areas. The clusters identified in these departments enclosed more than one farm with BBTD. For banana plots that were up to 5 years old, the preceding crop influenced the associated risk of BBTD infestation. BBTD could have a more complex natural history than simple vector-borne transmission.

It is estimated that less than 5% of the seed planted with bananas in these communes comes from certified seed sources. Without an adequate seed certification system, informal trade in seed bananas might provide a key pathway for long-distance virus dissemination, expanding significantly the range of virus dispersion, and becoming a major driver of disease propagation. Short-distance dispersion (secondary infections) are a result of aphid movement. After the establishment of the infection, vector transmission could then become an efficient short-distance mechanism of virus dispersion to other plants within the farm and along with the surrounding neighbouring crops.

Nevertheless, improvements could be made to the current model. A possible extension of this method to the Spatio-temporal framework is envisaged: it is sufficient to define an appropriate period between events. This could constitute an alternative to space-time scanning statistics hence improve the temporal response to accommodate management activities in disease modelling. Moreover, using UAVs equipped with a spectral and thermal camera to identify disease plants [23] and to take direct measurements could be possible in the next study and landscape scale to improve the accuracy of vegetation mapping by including plant stress indicators.

## Supporting information

**S1 Fig. Orthomosaic photo for a field in Dangbo.** (Red colour) Bananas.
(TIF)

**S2 Fig. Banana cropping systems in Benin.** (Left) Open gardens. (Right) Backyard gardens.
(TIF)

**S3 Fig. The estimates of BBTD risk from field sampled variables.** (Red colour) High risk. (Gold colour) Medium risk. (Blue colour) Low risk. (Lightgrey colour) Not significant. (Darkgrey colour) Not detected.
(TIF)

**S4 Fig. Prevalence per field crop associated and BBTD count.**
(TIF)

**S5 Fig. Prevalence per field crop succession and BBTD count.**
(TIF)

**S6 Fig. Correlation between BBTD and continuous variables.** (Q13) Observed case of BBTD: response variable. (Q12) Density of banana mats on the plot. (Q11) Age of banana: planting date. (Q40.East) Distance from target field to nearby banana field in East. (Q40.West) Distance from target field to nearby banana field in West. (Q40.South) Distance from target field to nearby banana field in South. (Q40.North) Distance from target field to nearby banana field in North.
(TIF)

**S7 Fig. Proportion of gardens with aphids.**
(TIF)

**S8 Fig. Average BBTD cases among aphid present and absent gardens.**
(TIF)

**S9 Fig. Mean cases in gardens receiving and not receiving seed from neighbours.**
(TIF)

**S10 Fig. Case counts by cutting/rooting banana mats infected.**
(TIF)

**S11 Fig. Mean BBTD prevalence in garden distancing.** (Black colour) Average East. (White colour) Average West. (Light grey colour) Average South. (Dark grey colour) Average North.
(TIF)

**S12 Fig. Mean BBTD prevalence in garden neighbouring.** (Black colour) Average East. (White colour) Average West. (Light grey colour) Average South. (Dark grey colour) Average North.
(TIF)

**S13 Fig. Conclusive diagram.**
(TIF)

**S1 Table. Model estimation results of Poisson regression.**
(DOCX)

## Acknowledgments

We thank the banana production community groups in Akpro-Missérété, Adjarra, Dangbo, Sakété, Athiémé, and Houéyogbé, Benin, for their cooperation in the mapping activity. We thank Christian Allikponto, and Vodonou Marc, Joseph Hodonou, and Camel Sogninou for supporting field data collection. We are grateful for the support of Prof Adanhounmè Villévo, in helping with the study design, and to Prof Baloïtcha Ezinvi, Dr Miriam F. Karlsson, for reviewing earlier versions of the manuscript. This study was carried out under the Learning Alliance for BBTD Control in Africa Framework (www.learningalliance.com) supported by the RTB-CRP Flagship 2. We thank the RTB and its partners (www.rtb.org).

## Author Contributions

**Conceptualization:** Martine Zandjanakou Tachin, Bonaventure Aman Omondi.

**Formal analysis:** Kéladomé Maturin Géoffroy Dato, Mahougnon Robinson Dégbègni, Mintodê Nicodème Atchadé.

**Funding acquisition:** Bonaventure Aman Omondi.

**Investigation:** Bonaventure Aman Omondi.

**Methodology:** Kéladomé Maturin Géoffroy Dato, Mintodê Nicodème Atchadé, Mahouton Norbert Hounkonnou, Bonaventure Aman Omondi.

**Project administration:** Bonaventure Aman Omondi.

**Software:** Kéladomé Maturin Géoffroy Dato.

**Supervision:** Mintodê Nicodème Atchadé, Mahouton Norbert Hounkonnou, Bonaventure Aman Omondi.

**Validation:** Mahouton Norbert Hounkonnou, Bonaventure Aman Omondi.

**Visualization:** Mahougnon Robinson Dégbègni, Martine Zandjanakou Tachin, Bonaventure Aman Omondi.

**Writing – original draft:** Kéladomé Maturin Géoffroy Dato, Bonaventure Aman Omondi.

**Writing – review & editing:** Kéladomé Maturin Géoffroy Dato.

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
