## [Decision Letter · Decision Letter 0]

15 Sep 2021

PONE-D-21-23976Spatial parameters associated with the risk of banana bunchy top disease in smallholder systemsPLOS ONE

Dear Dr. Géoffroy 

Thank you for submitting your manuscript to PLOS ONE. After careful consideration, we feel that it has merit but does not fully meet PLOS ONE’s publication criteria as it currently stands. Therefore, we invite you to submit a revised version of the manuscript that addresses the points raised during the review process. 

We look forward to receiving your revised manuscript.

Kind regards,

Tunira Bhadauria, Ph.D.

Academic Editor

PLOS ONE

2. Please provide in the Methods section details of how the human participants provided consent, and whether this was written or verbal. Please also clarify who granted permission for the UAV and ground surveys

3. Please include your tables as part of your main manuscript and remove the individual files. Please note that supplementary tables (should remain/ be uploaded) as separate "supporting information" files

“This work was funded by the University of Queensland Grant to Bioversity International under the project: BBTV mitigation: Community management in Nigeria and screening wild banana progenitors for resistance – Remote Mapping and Detection of Banana Canopy in Mixed Landscapes (Number 1229). Community Management of BBTD was funded by the Consortium Research Programme on Root Tubers and Bananas, BA 3.4, Banana Viral Diseases.”

“We thank the banana production community groups in Akpro-Missérété, Adjarra, Dangbo, Sakété, Athiémé, and Houéyogbé, Benin, for their cooperation in the mapping activity. We also have this work was funded by the University of Queensland Grant to Bioversity International under the project: BBTV mitigation: Community management in Nigeria and screening wild banana progenitors for resistance – Remote Mapping and Detection of Banana Canopy in Mixed Landscapes (Number 1229). Community Management of BBTD was funded by the Consortium Research Programme on Root Tubers and Bananas, BA 3.4, Banana Viral Diseases. We are grateful to RTB and her donor consortium.”

We note that you have provided information within the Acknowledgements Section. Please note that funding information should not appear in the Acknowledgments section or other areas of your manuscript. We will only publish funding information present in the Funding Statement section of the online submission form.

“This work was funded by the University of Queensland Grant to Bioversity International under the project: BBTV mitigation: Community management in Nigeria and screening wild banana progenitors for resistance – Remote Mapping and Detection of Banana Canopy in Mixed Landscapes (Number 1229). Community Management of BBTD was funded by the Consortium Research Programme on Root Tubers and Bananas, BA 3.4, Banana Viral Diseases”

6. We note that you have indicated that data from this study are available upon request. PLOS only allows data to be available upon request if there are legal or ethical restrictions on sharing data publicly. For more information on unacceptable data access restrictions, please see http://journals.plos.org/plosone/s/data-availability#loc-unacceptable-data-access-restrictions.

7. We note that Figure 1 in your submission contain maps images which may be copyrighted. All PLOS content is published under the Creative Commons Attribution License (CC BY 4.0), which means that the manuscript, images, and Supporting Information files will be freely available online, and any third party is permitted to access, download, copy, distribute, and use these materials in any way, even commercially, with proper attribution. For these reasons, we cannot publish previously copyrighted maps or satellite images created using proprietary data, such as Google software (Google Maps, Street View, and Earth). For more information, see our copyright guidelines: http://journals.plos.org/plosone/s/licenses-and-copyright.

 a. You may seek permission from the original copyright holder of Figure(s) [#] to publish the content specifically under the CC BY 4.0 license. 

8. Please include a separate caption for each figure in your manuscript

Additional Editor Comments (if provided):

After going through the manuscript I agree with the reviewer number one that though the manuscript is well written with enough scientific input to be considered for publication in the journal,but still there are some points which authors need to pay attention and respond to before the manuscript can be accepted for the publication.

The writers must react to the reviewers' concerns and integrate their suggestions into the updated work. As a result, modest revisions are recommended before the paper is accepted for publication.

Reviewers' comments:

Reviewer's Responses to Questions

**Comments to the Author**

1. Is the manuscript technically sound, and do the data support the conclusions?

Reviewer #1: Yes

Reviewer #2: Yes

2. Has the statistical analysis been performed appropriately and rigorously? 

Reviewer #1: Yes

Reviewer #2: Yes

3. Have the authors made all data underlying the findings in their manuscript fully available?

Reviewer #1: Yes

Reviewer #2: Yes

4. Is the manuscript presented in an intelligible fashion and written in standard English?

Reviewer #1: Yes

Reviewer #2: Yes

5. Review Comments to the Author

Reviewer #1: Author are suggested folowing improvement in manuscript

1.In abstract"Mapping was done in This study area ..."This must be like this instead of This.

2.Author are requested to improve english Like " but cooking banana varieties exist also" in introductiion pl reframe centance.

3.Author are requested to provide key words and explanation in abstract and introduction about main objective of this manuscript like Studies to understand BBTD epidemiology etc.

4."Overall, our data show that crop succession, production, longevity, and informal seed systems to be associated with a higher risk of BBTD; while mixed cropping matrices, higher altitude, and lower surrounding BBTD pressure were linked to reduced BBTD prevalence" appeling conclusion.Pl mention it as a question in abstract and introduction and later as you provided answer and explanation .Its remarkable and very interesting studies.

5.Author requested to draw a conclusive diagram of the study which may represent the best strategy for combating BBTD issue.

Beside biotechnology and molecular breeding programme ,management of vector and crop doiversification and such type of model are really very useful.Authors did very interersting work but conclusion is not apeling.Author must improve manuscript by introducing key word,key questions and key answer of the problem.

Reviewer #2: Manuscript has been written in well scientific manner. I did not found any major correction in the MS. Although, some minor corrections are needed.

• Introduction and discussion is too large.

• Please specify the future perspective of your study.

• Why your study is important for society, elaborate.

6. PLOS authors have the option to publish the peer review history of their article (what does this mean?). If published, this will include your full peer review and any attached files.

Reviewer #1: **Yes: **Kuldip Jayaswall

Reviewer #2: No

---

## [Author Response · Author response to Decision Letter 0]

15 Nov 2021

Thanks dear reviewers.

Please found the responses in the file names ''Response to reviwers''

---

## [Editor Report · Decision Letter 1]

22 Nov 2021

Spatial parameters associated with the risk of banana bunchy top disease in smallholder systems

PONE-D-21-23976R1

Dear Dr. Géoffroy DATO

We’re pleased to inform you that your manuscript has been judged scientifically suitable for publication and will be formally accepted for publication once it meets all outstanding technical requirements.

Kind regards,

Tuneera Bhadauria, Ph.D.

Academic Editor

PLOS ONE

Additional Editor Comments (optional):

I'd want to congratulate the authors on updating the paper in response to the reviewers' concerns, individually responding to each comment and incorporating it into the text when appropriate. The manuscript meets the journal's scientific requirements for publication. As a result, I recommend that the manuscript be accepted for publication in the journal.
---

## [Editor Report · Acceptance letter]

26 Nov 2021

PONE-D-21-23976R1 

Spatial parameters associated with the risk of banana bunchy top disease in smallholder systems 

Dear Dr. DATO:

I'm pleased to inform you that your manuscript has been deemed suitable for publication in PLOS ONE. Congratulations! Your manuscript is now with our production department. 

Kind regards, 

on behalf of

Dr. Tunira Bhadauria 

Academic Editor

PLOS ONE